# `Quadproj`: a Python package for projecting onto quadratic hypersurfaces

## Abstract

Quadratic hypersurfaces are a natural generalization of affine subspaces, and projections are elementary blocks of algorithms in optimization and machine learning. It is therefore intriguing that no proper studies and tools have been developed to tackle this nonconvex optimization problem. The `quadproj` package is a user-friendly and documented software that is dedicated to project a point onto a non-cylindrical central quadratic hypersurface.

## 1 Introduction

Projection is one of the building blocks in many optimization softwares and machine learning algorithms [7, §2.9]. Projection applications are multiple and include projected (gradient) methods [9, 19], alternating projections [12, 11], splitting methods [13], and other proximal methods [18].

In this work, we focus on the orthogonal projection onto a quadratic surface. The motivation is threefold. First, quadratic (hyper)surfaces are a natural generalization of affine subspaces. Because the projection onto an affine subspace is easy, it is tempting to trade accurate representation of the subspace (*i.e.*, by approximating the quadratic hypersurface as a hyperplane) so as to benefit from an easiest projection, see [17] for an example of this kind. Being able to easily project onto a quadratic hypersurface, or *quadric*, would remove the need of this trade-off. Second, the projection onto a quadratic hypersurface is a direct requirement of some applications: either in 2D and 3D spaces (mostly in image processing and computer-aided design) [14, 24, 10], or in larger dimensional spaces such as the nonconvex economic dispatch [22], the security of the gas network [20], and local learning methods [6]. Finally, being able to project onto a quadratic hypersurface can be seen as the first step to project onto the intersection of quadratic hypersurfaces. And, it is a classical result of algebraic geometry that any projective variety is isomorphic to an intersection of quadratic hypersurfaces [8, Exercise 2.9].

We implement the method proposed in [22] and package it into a Python library. This method consists in solving the nonlinear system of equations associated to the KKT conditions of the nonlinear optimization problem used to define the projection. To alleviate the complexity increase with the size of the problem (because the number of critical points grows linearly with the size of the problem), the authors of [22] show that one of the global minima, that is, one of the projections, either corresponds to the unique root of a nonlinear univariate function on a known interval, or belongs to a finite set of points to which a closed-form is available. The root of the univariate solution is readily obtained *via* Newton's method. Hence, the bottleneck of this method is the eigendecomposition of the matrix that is used to define the quadric.

A few other studies also discuss the projection onto quadrics. For the 2D or 3D cases, some methods are discussed in [15, 14, 10], but they do not present the extension to the $n$-dimensional case. The

35 $n$-dimensional case is also analyzed in [21], but their method is an iterative scheme that may converge
36 slowly and sometimes fails to provide the exact projection.

37 The main goal of the present study is to democratize the *exact* method from [22], and thereby to save
38 any potential user of a quadratic projection from implementing it (or from falling back to approximate
39 the quadratic hypersurface by a hyperplane). Hence, emphasis is placed on i) the ease of installation
40 and ii) the user-friendliness of the package.

41 The package is available in the Python Package Index (PyPi) [3] and on conda [2]. The source code
42 is open-sourced on GitLab [4] and the documentation is available in [1].

## 2 Problem formulation

44 In this section, we first shortly present the projection problem. Then, we define the feasible set onto
45 which the projection is performed (*i.e.*, a non-cylindrical central quadric).

### 2.1 The projection problem

47 The projection problem consists in mapping a point $\boldsymbol{x}^0$ onto a subset $C$ of some Hilbert space $H$,
48 while minimizing the distance $\|\cdot\|_H$ that is induced by the inner product $\langle\cdot,\cdot\rangle_H$:

$$\mathrm{Pr}_C\left(\boldsymbol{x}\right) = \arg\min_{\boldsymbol{x}\in C}\|\boldsymbol{x}-\boldsymbol{x}^0\|_H.$$

49 For nonempty closed sets $C$ the projection is nonempty [22, Prop. 2.1]. It is a singleton *if* $C$ is also
50 convex. For a nonconvex closed set $C$, the solution may be a singleton (*e.g.*, $\mathrm{Pr}_C\left(\boldsymbol{x}^0\right)$ with $\boldsymbol{x}^0 \in C$),
51 a larger finite set (*e.g.*, the projection of any point that lies at mid distance between two hyperplanes
52 onto the set defined by the union of these two hyperplanes), or an infinite set (*e.g.*, the projection of
53 the center of a sphere onto the sphere itself).

54 In the case where $C$ is a hyperplane, there exists a closed-form solution. If, for some vector $\boldsymbol{b} \in H$,
55 we have

$$C = \left\{\boldsymbol{x} \in H \,\middle|\, \langle\boldsymbol{b},\boldsymbol{x}\rangle_H + c = 0\right\},$$

56 then the projection is the following singleton:

$$\mathrm{Pr}_C\left(\boldsymbol{x}^0\right) = \left\{\boldsymbol{x}^0 - \frac{\langle\boldsymbol{b},\boldsymbol{x}^0\rangle_H + c}{\|\boldsymbol{b}\|_H}\boldsymbol{b}\right\}.$$

57 In this paper, we consider the canonical $n$-dimensional Hilbert space $H = \mathbb{R}^n$ equipped with the
58 canonical inner product ($\langle\boldsymbol{u},\boldsymbol{v}\rangle_H = \boldsymbol{u}^\intercal\boldsymbol{v}$) and its induced norm ($\|\boldsymbol{u}\|_H = \|\boldsymbol{u}\|_2 = \sqrt{\boldsymbol{u}^\intercal\boldsymbol{u}}$).

59 In this settings, we present a toolbox for computing the projection onto a non-cylindrical central
60 quadric.

### 2.2 Non-cylindrical central quadrics

62 A *quadric* $\mathcal{Q}$ is the generalization of conic sections in spaces of dimension larger than two. It is a
63 quadratic hypersurface of $\mathbb{R}^n$ (of dimension $n-1$) that can be characterized as

$$\mathcal{Q} = \left\{\boldsymbol{x} \in \mathbb{R}^n \,\middle|\, \Psi(\boldsymbol{x}) := \boldsymbol{x}^\intercal\boldsymbol{A}\boldsymbol{x} + \boldsymbol{b}^\intercal\boldsymbol{x} + c = 0\right\}, \tag{1}$$

64 with $\boldsymbol{A} \in \mathbb{R}^{n\times n}$ a symmetric matrix, $\boldsymbol{b} \in \mathbb{R}^n$, $c \in \mathbb{R}$, and $\Psi(\boldsymbol{x})\colon \mathbb{R}^n \to \mathbb{R}$ a nonzero quadratic
65 function.

66 We can also represent the quadric with the extended coordinate vector $\boldsymbol{x}^* \in \mathbb{R}^{n+1}$ by inserting 1 in
67 the first row of the coordinate $\boldsymbol{x}$. Using the *extended* (symmetric) *matrix*

$$\boldsymbol{A}^* := \left(\begin{array}{c|c} c & \boldsymbol{b}^\intercal/2 \\ \hline \boldsymbol{b}/2 & \boldsymbol{A} \end{array}\right), \tag{2}$$

the quadric is equally defined as

$$\mathcal{Q} = \left\{ \boldsymbol{x} = \begin{pmatrix} x_1 \\ \vdots \\ x_n \end{pmatrix} \in \mathbb{R}^n \,\middle|\, (1 \quad x_1 \quad \dots \quad x_n) \, \boldsymbol{A}^* \begin{pmatrix} 1 \\ x_1 \\ \vdots \\ x_n \end{pmatrix} = 0 \right\}.$$

Let $r$ be the rank of $\boldsymbol{A}$ (denoted as $\mathrm{rk}(A)$) and $p$ be the number of positive eigenvalues of $\boldsymbol{A}$. Following the classification of [16, Theorem 3.1.1], we distinguish three types of real quadrics.

- Type 1, **conical** quadrics: $0 \le p \le r \le n, p \ge r - p, \mathrm{rk}(\boldsymbol{A}^*) = \mathrm{rk}(\boldsymbol{A}|\frac{\boldsymbol{b}}{2}) = r$.
- Type 2, **central** quadrics: $0 \le p \le r \le n, \mathrm{rk}(\boldsymbol{A}^*) > \mathrm{rk}(\boldsymbol{A}|\frac{\boldsymbol{b}}{2}) = r$.
- Type 3, **parabolic** quadrics: $0 \le p \le r < n, \mathrm{rk}(\boldsymbol{A}|\frac{\boldsymbol{b}}{2}) > r$.

We also call **cylindrical** quadrics the central and conical quadrics with $r < n$ and the parabolic quadrics with $r < n - 1$.

In this paper, we focus on nonempty **central** and **non-cylindrical** quadrics, that is, we consider Eq. (1) with $\boldsymbol{A}$ nonsingular and $c \ne \frac{\boldsymbol{b}^\mathsf{T} \boldsymbol{A}^{-1} \boldsymbol{b}}{4}$. Indeed, when $\boldsymbol{A}$ is nonsingular (*i.e.*, when $r = n$), one can show that the condition $c \ne \frac{\boldsymbol{b}^\mathsf{T} \boldsymbol{A}^{-1} \boldsymbol{b}}{4}$ is equivalent to $\mathrm{rk}(\boldsymbol{A}^*) > \mathrm{rk}(\boldsymbol{A}|\frac{\boldsymbol{b}}{2})$, see [23, § 2.5] for more details.

Note that *central* quadrics are characterized by the existence of a center $\boldsymbol{d} = -\frac{\boldsymbol{A}^{-1}\boldsymbol{b}}{2}$, which corresponds to the center of symmetry of the quadric.

In 2D, a non-cylindrical central quadric can be a circle, an ellipse, or a hyperbola. In 3D, it can be a sphere, an ellipsoid, a one-sheet hyperboloid, or a two-sheet hyperboloid. In higher dimensional spaces, we have hyperspheres, (hyper)ellipsoids, and hyperboloids.

## 2.3 The projection as an optimization problem

Let $\tilde{\boldsymbol{x}}^0 \in \mathbb{R}^n$ be the point to be projected, and $\mathcal{Q}$ be a non-cylindrical central quadric with parameters $\boldsymbol{A}, \boldsymbol{b}$, and $c$. The optimization problem at hand reads

$$\min_{\tilde{\boldsymbol{x}} \in \mathbb{R}^n} \|\tilde{\boldsymbol{x}} - \tilde{\boldsymbol{x}}^0\|_2$$
$$\text{subject to } \tilde{\boldsymbol{x}}^\mathsf{T} \boldsymbol{A} \tilde{\boldsymbol{x}} + \boldsymbol{b}^\mathsf{T} \tilde{\boldsymbol{x}} + c = 0. \tag{3}$$

Using an appropriate coordinate transformation, we can simplify Eq. (3). Let $\boldsymbol{V} \boldsymbol{D} \boldsymbol{V}^\mathsf{T} = \boldsymbol{A}$ be an eigendecomposition of $\boldsymbol{A}$, with $\boldsymbol{V} \in \mathbb{R}^{n \times n}$ an orthogonal matrix whose columns are eigenvectors of $\boldsymbol{A}$ and $\boldsymbol{D} = \mathrm{diag}(\boldsymbol{\lambda})$ the diagonal matrix whose entries are the associated eigenvalues of $\boldsymbol{A}$ (denoted as $\boldsymbol{\lambda}$ and sorted in descending order), and let $\gamma = c + \boldsymbol{b}^\mathsf{T} \boldsymbol{d} + \boldsymbol{d}^\mathsf{T} \boldsymbol{A} \boldsymbol{d} = c - \frac{\boldsymbol{b}^\mathsf{T} \boldsymbol{A}^{-1} \boldsymbol{b}}{4}$.

We can guarantee that $\gamma > 0$ by flipping, if needed, the sign of $\boldsymbol{A}, \boldsymbol{b}$, and $c$. Indeed, $\boldsymbol{x} \in \mathcal{Q} \Leftrightarrow \boldsymbol{x}^\mathsf{T} \boldsymbol{A} \boldsymbol{x} + \boldsymbol{b}^\mathsf{T} \boldsymbol{x} + c = 0 \Leftrightarrow \boldsymbol{x}^\mathsf{T}(-\boldsymbol{A})\boldsymbol{x} + (-\boldsymbol{b})^\mathsf{T} \boldsymbol{x} + (-c) = 0$, but if $\gamma = c - \frac{-\boldsymbol{b}^\mathsf{T} \boldsymbol{A}^{-1}\boldsymbol{b}}{4} < 0$, then $(-c) - \frac{(-\boldsymbol{b}^\mathsf{T})(-\boldsymbol{A}^{-1})(-\boldsymbol{b})}{4} = -\gamma > 0$.

If we define the linear transformation

$$T \colon \mathbb{R}^n \to \mathbb{R}^n \colon \tilde{\boldsymbol{x}} \mapsto T(\tilde{\boldsymbol{x}}) = \boldsymbol{V}^\mathsf{T} \frac{(\tilde{\boldsymbol{x}} - \boldsymbol{d})}{\sqrt{\gamma}}, \tag{4}$$

then $Eq.$ (3) can be rewritten as

$$\min_{\boldsymbol{x} \in \mathbb{R}^n} \|\boldsymbol{x} - \boldsymbol{x}^0\|_2^2$$
$$\text{subject to } \sum_{i=1}^n \lambda_i x_i^2 - 1 = 0, \tag{5}$$

with $\boldsymbol{x}^0 = T(\tilde{\boldsymbol{x}}^0)$. Note that $\sum_{i=1}^n \lambda_i x_i^2 = \boldsymbol{x}^\mathsf{T} \boldsymbol{D} \boldsymbol{x}$, and that in this new coordinate system the quadric is centered at the origin and aligned with the axes.

## 3 Method

There exists at least one global solution of Eq. (5) because the objective function is a real-valued, continuous and coercive function defined on a nonempty closed set. Let us characterize one of these solutions.

The Lagrangian function of Eq. (5), with Lagrange multiplier $\mu$ and with $\boldsymbol{D} = \mathrm{diag}(\boldsymbol{\lambda}) \in \mathbb{R}^{n \times n}$, reads

$$\mathcal{L}(\boldsymbol{x}, \mu) = (\boldsymbol{x} - \boldsymbol{x}^0)^{\mathsf{T}}(\boldsymbol{x} - \boldsymbol{x}^0) + \mu(\boldsymbol{x}^{\mathsf{T}}\boldsymbol{D}\boldsymbol{x} - 1). \tag{6}$$

Because the center does not belong to the quadric, the linear independence constraint qualification (LICQ) criterion is satisfied; using the KKT conditions, we have that any solution of Eq. (5) must be a solution of the following system of nonlinear equations [5, Chapter 4]:

$$\boldsymbol{\nabla}\mathcal{L}(\boldsymbol{x}, \mu) = \begin{pmatrix} 2(\boldsymbol{x} - \boldsymbol{x}^0) + 2\mu\boldsymbol{D}\boldsymbol{x} \\ \boldsymbol{x}^{\mathsf{T}}\boldsymbol{D}\boldsymbol{x} \end{pmatrix} = \boldsymbol{0}. \tag{7}$$

For $\mu \notin \pi(\boldsymbol{A}) := \left\{ -\frac{1}{\lambda} \mid \lambda \text{ is an eigenvalue of } \boldsymbol{A} \right\}$, we write the $n$ first equations of Eq. (7) as

$$\boldsymbol{x}(\mu) = (\boldsymbol{I} + \mu\boldsymbol{D})^{-1}\boldsymbol{x}^0. \tag{8}$$

Injecting this expression in the last equation of Eq. (7), we obtain a univariate and extended-real valued function

$$f \colon \mathbb{R} \to \overline{\mathbb{R}} \colon \mu \mapsto f(\mu) = \boldsymbol{x}(\mu)^{\mathsf{T}}\boldsymbol{D}\boldsymbol{x}(\mu) - 1$$
$$= \sum_{i=1, x_i^0 \neq 0}^{n} \lambda_i \left( \frac{x_i^0}{1 + \mu\lambda_i} \right)^2 - 1. \tag{9}$$

And any root of $f$ corresponds to a KKT point.

In [22, Proposition 2.20], the authors show that there is an optimal solution of Eq. (5) in the set $\{\boldsymbol{x}(\mu^*)\} \bigcup \boldsymbol{X}^d$ where

- $\boldsymbol{x}(\mu)$ is defined by Eq. (8), $\mu^*$ is the unique root of $f$ on a given open interval $\mathcal{I}$;
- $\boldsymbol{X}^{\mathrm{d}}$ is a finite set of less than $n$ elements.

The set $\boldsymbol{X}^{\mathrm{d}}$ is nonempty only if $\tilde{\boldsymbol{x}}^0$ is located on at least one principal axis of the quadric (or equivalently, if at least one entry of $\boldsymbol{x}^0$ is 0), we refer to such cases as *degenerate cases* (examples of which are depicted in Fig. 4). The details and the explicit formulation of $\mathcal{I}$ and $\boldsymbol{X}^{\mathrm{d}}$ are given in [22, § 2.5].

Our strategy to solve Eq. (5) is to compute all elements of $\boldsymbol{X}^{\mathrm{d}}$ and the root of $f$ on $\mathcal{I}$, and to choose among these points the one that is the closest to $\boldsymbol{x}^0$. We can then return the optimal solution of Eq. (3) by using the inverse transformation

$$T^{-1} \colon \mathbb{R}^n \to \mathbb{R}^n \colon \boldsymbol{x} \mapsto T^{-1}(\boldsymbol{x}) = \sqrt{\gamma}\,\boldsymbol{V}\boldsymbol{x} + \boldsymbol{d}. \tag{10}$$

We denote the (unique) returned solution as $\mathrm{Pr}_{\mathcal{Q}}(\boldsymbol{x})$, which is *one* of the optimal solutions of Eq. (5).

The root of $f$ is effectively obtained with Newton's method, which benefits from a superlinear convergence. Moreover, the number of iterations—which amounts to evaluating $f$ and $f'$ for a cost $\mathcal{O}(n)$—is typically low (no more than 20) and is independent from $n$. The computation of the finite set $\boldsymbol{X}^{\mathrm{d}}$ also costs $\mathcal{O}(n)$. These computations are negligible with respect to the eigendecomposition, which is the bottleneck of the method. In particular, for 100 problems of size $n = 500$, we obtain a mean execution time of $0.065\,\mathrm{s}$ for the root-finding algorithm and a mean execution time of $0.66\,\mathrm{s}$ for the eigendecomposition (this experiment is available in `test_newton.py` in [4]).

Another method for solving Eq. (3) (while trying to avoid the computation of the eigendecomposition of $\boldsymbol{A}$) is to compute the gradient of the Lagrangian of Eq. (3) and to use a dedicated solver of systems of nonlinear equations. In this paper, we use the method `optimize.fsolve` from the python package `scipy`. In Fig. 1, we observe that for dimensions larger than 100, `quadproj` is faster than `fsolve`; each data point in Fig. 1 is the mean of 10 randomly generated instances, and the code of this experiment is available in `test_execution_time.py` in [4]. Besides, it is not guaranteed

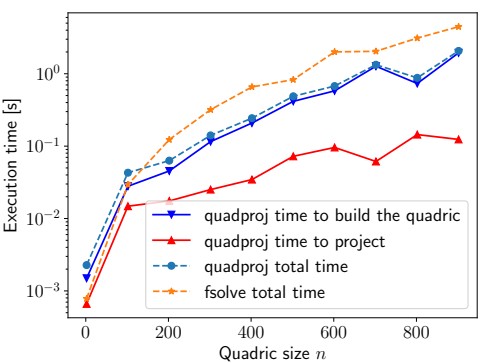
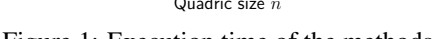
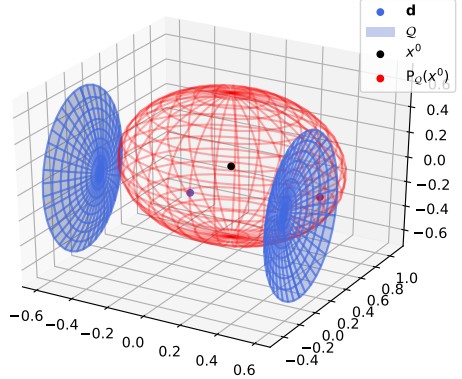

Figure 1: Execution time of the methods.

Figure 2: Output of listing 8.

that `fsolve` returns the correct root (*i.e.*, it may converge to a critical point of Eq. (3) that is not the global minimizer) nor that it will converge at all. Finally, `fsolve` cannot detect the additional solutions that appear in the degenerate cases; identifying that the case is degenerate requires the eigendecomposition of $A$ which would upsurge the execution time of such an `fsolve`-based method. For all these reasons, we decided not to make this `fsolve`-based method available in the `quadproj` package.

# 4   The `quadproj` package

Let us demonstrate in this section the use of `quadproj` through small code snippets. To avoid redundancy (*e.g.*, in the imports), the snippets should be run in the current order.

## 4.1   The basics: a simple $n$-dimensional example

In listing 1, we create in line 16 an object of class `quadproj.quadrics.Quadric` obtained by providing a `dict` (param) that contains the entries `'A'`, `'b'`, and `'c'` (corresponding to the parameters $A$, $b$, and $c$). We then create a random initial point x0, project it onto the quadric, and check that the resulting point x_project is feasible by using the instance method `Quadric.is_feasible`.

Listing 1: Projection onto a $n$-dimensional quadric.

```python
from quadproj import quadrics
from quadproj.project import project

import numpy as np

# creating random data
dim = 42
_A = np.random.rand(dim, dim)
A = _A + _A.T  # make sure that A is symmetric
b = np.random.rand(dim)
c = -1.42

param = {'A': A, 'b': b, 'c': c}
Q = quadrics.Quadric(param)

x0 = np.random.rand(dim)
x_project = project(Q, x0)
assert Q.is_feasible(x_project), 'The projection is incorrect!'
```

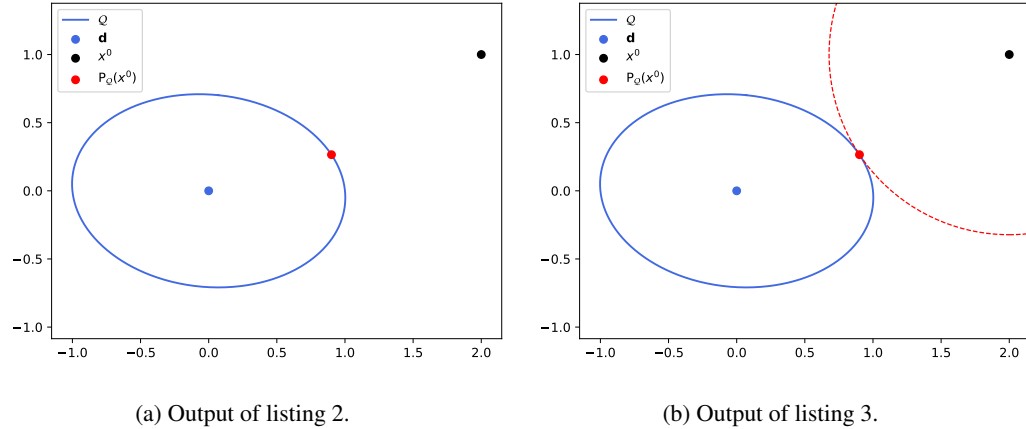

(a) Output of listing 2.

(b) Output of listing 3.

Figure 3: Projection onto an ellipse.

## 4.2 Visualise the solution

The package also provides visualization tools. In listing 2, we compute and plot the projection of a point onto an ellipse. The output is given in Fig. 3a where the projection `x_project` of x0 onto the quadric is depicted as a red point.

Listing 2: 2D visualization.

```
from quadproj.project import plot_x0_x_project
from os.path import join

import matplotlib.pyplot as plt

output_path = '../images/'

show = False

A = np.array([[1, 0.1], [0.1, 2]])
b = np.zeros(2)
c = -1
Q = quadrics.Quadric({'A': A, 'b': b, 'c': c})

x0 = np.array([2, 1])
x_project = project(Q, x0)

fig, ax = Q.plot(show=show)
plot_x0_x_project(ax, Q, x0, x_project)
# ax.axis('equal')
plt.savefig(output_path, 'ellipse_no_circle.pdf'))
```

A quick glance at Fig. 3a might give the (false) impression that the red point is *not* the closest one: this is due to the difference in scale between both axes. As a way to remedy this issue, we can either impose equal axes (by uncommenting line 20 in listing 2) or setting the argument `flag_circle=True`. The latter plots a circle centred in $x^0$ with radius $\|x^0 - \mathrm{Pr}_{\mathcal{Q}}\left(x^0\right)\|_2$. Because of the difference in the axis scaling, this circle (Fig. 3b) might resemble an ellipse. However, it should not cross the quadric and be tangent to the quadric at $\mathrm{Pr}_{\mathcal{Q}}\left(x^0\right)$; this is a visual proof of the solution optimality.

Listing 3: 2D visual proof of the optimality.

```
fig, ax = Q.plot()
plot_x0_x_project(ax, Q, x0, x_project, flag_circle=True)
fig.savefig(join(output_path, 'ellipse_circle.pdf'))
```

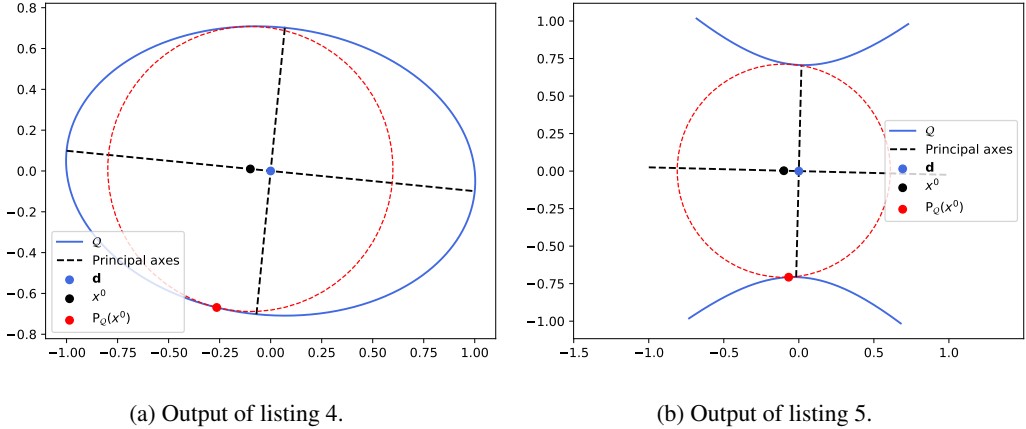

(a) Output of listing 4.                    (b) Output of listing 5.

Figure 4: Degenerate projections.

## 4.3  Degenerate cases

For constructing a degenerate case, we can:

- Either construct a quadric in standard form, *i.e.*, with a diagonal matrix A, a nul vector b, c=-1 and define some x0 with a least one entry equal to zero;
- Or choose any quadric and select x0 to be on any principal axis of the quadric.

Let us illustrate the second option in listing 4. We create x0 by applying the (inverse) standardization (see, Eq. (10)) from some x0 with at least one entry equal to zero.

Here, we chose to be close to the centre and on the longest axis of the ellipse so as to be sure that there are multiple (two) solutions.

Recall that the program returns *only one solution*. Multiple solutions is planned in future releases.

Listing 4: Degenerate projection onto an ellipse.

```
1 x0 = Q.to_non_standardized(np.array([0, 0.1]))
2 x_project = project(Q, x0)
3 fig, ax = Q.plot(show_principal_axes=True)
4 ax.legend(loc='lower left')
5 plot_x0_x_project(ax, Q, x0, x_project, flag_circle=True)
6 fig.savefig(join(output_path, 'ellipse_degenerated.pdf'))
```

The output figure `ellipse_degenerated.pdf` is given in Fig. 4a. It can be seen that the reflection of `x_project` along the largest ellipse axis (visible because `show_principal_axes=True`) yields another optimal solution.

## 4.4  Supported quadrics

The class of supported quadrics are the non-cylindrical central quadrics. Visualization tools are available for the 2D and 3D cases: ellipses, hyperbolas, ellipsoids and hyperboloids.

### 4.4.1  Ellipses

See previous section for examples of projection onto ellipses.

### 4.4.2  Hyperbolas

We illustrate in listing 5 the code to compute a (degenerated) projection onto a hyperbola. The figure output is depicted in Fig. 4b.

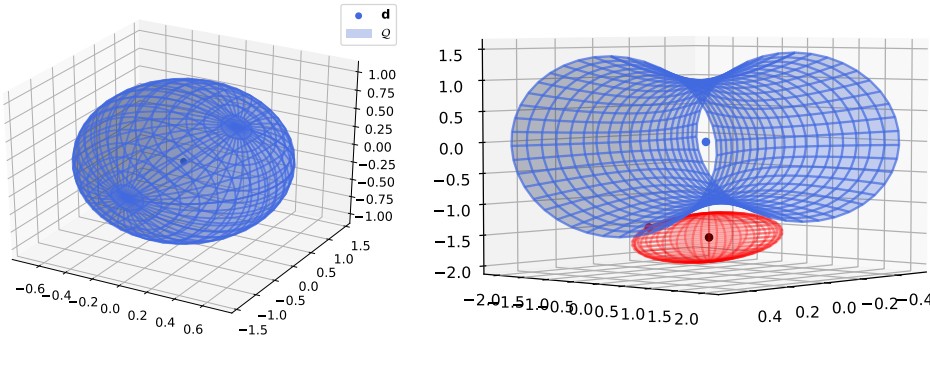

(a) Output of listing 6.  (b) Output of listing 7.

Figure 5: Visualizations of 3D quadrics.

In this case, there is no root to the nonlinear function $f$ from Eq. (9): graphically, the second axis
does not intersect the hyperbola. This is not an issue because two solutions are obtained from the
other set of KKT points ($\boldsymbol{X}^{\mathrm{d}}$).

Listing 5: Degenerate projection onto a hyperbola.

```
A[0, 0] = -2
Q = quadrics.Quadric({'A': A, 'b': b, 'c': c})
x0 = Q.to_non_standardized(np.array([0, 0.1]))
x_project = project(Q, x0)
fig, ax = Q.plot(show_principal_axes=True)
plot_x0_x_project(ax, Q, x0, x_project, flag_circle=True)
fig.savefig(join(output_path, 'hyperbola_degenerated.pdf'))
```

### 4.4.3 Ellipsoids

Similarly as the 2D case, we can plot an ellipsoid (listing 6) as in Fig. 5a. To ease visualization, the
function `get_turning_gif` lets you write a rotating gif.

Listing 6: Nondegenerate projection onto a one-sheet hyperboloid.

```
dim = 3
A = np.eye(dim)
A[0, 0] = 2
A[1, 1] = 0.5

b = np.zeros(dim)
c = -1
param = {'A': A, 'b': b, 'c': c}
Q = quadrics.Quadric(param)

fig, ax = Q.plot()

fig.savefig(join(output_path, 'ellipsoid.pdf'))

Q.get_turning_gif(step=4, gif_path=join(output_path, Q.type+'.gif'))
```

### 4.4.4 One-sheet hyperboloid

In listing 7, we illustrate the case of a one-sheet hyperboloid. Because it is currently not possible
to use equal axes in 3D plots with `matplotlib`, the `flag_circle` argument allows to confirm the
optimality of the solution despite the difference in the axis scales.

```
265  1  A[0, 0] = -4
266  2
267  3  param = {'A': A, 'b': b, 'c': c}
268  4  Q = quadrics.Quadric(param)
269  5
270  6  x0 = np.array([0.1, 0.42, -1.5])
271  7
272  8  x_project = project(Q, x0)
273  9
274 10  fig, ax = Q.plot()
275 11  plot_x0_x_project(ax, Q, x0, x_project, flag_circle=True)
276 12  ax.get_legend().remove()
277 13  ax.view_init(elev=4, azim=42)
278 14
279 15  fig.savefig(join(output_path, 'hyperboloid_circle.pdf'), bbox_inches='
280       tight')
```

### 4.4.5   Two-sheet hyperboloid

Finally, let us project a point onto a two-sheet hyperboloid: a quadratic surface with two positive eigenvalues and one negative eigenvalue.

Listing 8 is the program that produces Fig. 2. This is a degenerate case with two optimal solutions; `quadproj` returns one of these solutions (the one of the first orthant located in the right sheet of the hyperboloid).

Listing 8: Degenerate projection onto a two-sheet hyperboloid.

```
287  1  A = np.eye(3)
288  2  A[0, 0] = 4
289  3  A[1, 1] = -2
290  4  A[2, 2] = -1
291  5  b = np.zeros(3)
292  6  c = -1
293  7  param = {'A': A, 'b': b, 'c': c}
294  8  Q = quadrics.Quadric(param)
295  9
296 10  x0 = np.array([0, 0.5, 0])
297 11
298 12  x_project = project(Q, x0)
299 13
300 14  fig, ax = Q.plot(show_principal_axes=True)
301 15  plot_x0_x_project(ax, Q, x0, x_project, flag_circle=True)
```

## 5   Conclusion

In this paper, we presented a toolbox, called `quadproj`, for projecting any point onto a non-cylindrical central quadric. The problem is written as a smooth nonlinear optimization problem and the solution is characterized through the KKT conditions.

We implemented and distributed this toolbox while focusing on the user-friendliness and the simplicity of installation. It is therefore possible to install it from multiple sources (Pypi, conda, or from sources), and the projection is readily computed in a few lines of code.

Further research includes the extension to cylindrical central quadrics, and more generally to conical and parabolic quadrics. Another research direction is to reduce the execution time of the algorithm by focusing on the bottleneck of the method (*i.e.*, the eigendecomposition of the symmetric matrix used to define the quadric).

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
