# OpenReview forum: "Quadproj: a Python package for projecting onto quadratic hypersurfaces"
_NeurIPS.cc/2022/Conference — NeurIPS 2022 Submitted_

### Official Review · Reviewer_PLhJ · 2022-06-14

**Rating:** 8
**Confidence:** 4
**Soundness:** 3 good
**Presentation:** 4 excellent
**Contribution:** 3 good

**Summary:**

Presents a library for computing projections onto a range of surfaces defined by Quadrics.

**Questions:**

N/A

**Limitations:**

This paper addresses a very narrow problem, and does not illustrate any concrete applications. I would have liked to see some demonstration of applications of the library.

**Strengths And Weaknesses:**

This paper does a good job of describing the problem it solves and illustrating examples of how the library can be used to solve instances of problems within the class of projection problems it considers. I am supportive of papers on software libraries, they are explicitly within the scope of NeurIPS and contribute directly to the ML community. I don't have any larger critiques, the examples are easy to understand and the visuals are simple and clear.

---

### Official Review · Reviewer_o4hv · 2022-06-29

**Rating:** 3
**Confidence:** 4
**Soundness:** 3 good
**Presentation:** 3 good
**Contribution:** 2 fair

**Summary:**

This paper is a short tutorial on a Python optimization package, which implements the method of the recent preprint [21], regarding the problem of projecting a given point of the Euclidean n-dimensional space onto a quadratic hypersurface.

**Questions:**

1. What is the highest ambient dimension that the package can handle? Could the authors please report some indicative running times?

2. How does this package compare to generic state-of-the-art homotopy continuation methods applied to the same problem?

**Limitations:**

Yes.

**Strengths And Weaknesses:**

Strength:

Certainly, being able to project efficiently and accurately onto a quadratic hypersurface is very important, and the present paper concerns a promising step in this direction. Here is an additional reason for the significance of this problem, not mentioned in the paper: It is a classical fact in algebraic geometry, that any variety defined by homogeneous equations is isomorphic, via the so-called Veronese map, to a variety defined by quadratic equations (e.g., see Exercise 2.9, page 25, in “Algebraic Geometry: A First Course” by Joe Harris). Thus being able to efficiently and accurately project onto a single quadratic hypersurface, is the beginning of being able to project onto any variety, since the latter can be viewed as an intersection of quadratic hypersurfaces.

Weakness:

While a tutorial presenting a Python package for projecting onto quadratic hypersurfaces is certainly nice to have, I am of the impression that NeurIPS is not the appropriate venue for it.  Instead, I would have loved to see a short version of [21], with experiments or a case study highlighting the importance of the problem for the machine learning and data science community, and comparing with the state-of-the-art homotopy continuation methods for the same problem.

Bibliographic remark: I would like to bring to the authors’ attention the following papers and citations therein, regarding the problem of projecting onto hypersurfaces and varieties more generally:
1. Draisma, Horobet, Ottaviani, Sturmfels, Thomas. “The Euclidean distance degree of an algebraic variety”. Foundations of Computational Mathematics.
2. Breiding, Sottile, Woodcock. “Euclidean distance degree and mixed volume”. Foundations of Computational Mathematics.

---

> ### Author Response · Authors · 2022-08-01
> **Response to O4hv**
>
> The reviewer proposed an additional reason for the significance of the problem. We decided to add this point to the paper (see highlighted text in the introduction).
>
> The reviewer raises the same questions as reviewer WidE regarding the relevance of a package presentation in NeurIPS. He then argues on the possibility of leveraging quadproj for comparison with state-of-the-art methods.
> This is exactly the argument that drove us to submit here: offering to this large forum and audience that is NeurIPS a new tool that can be used in a variety of situations.
>
> Finally, the reviewer asks two questions. The first one has already been addressed in the answer to reviewer 47QC.
>
> As far as the second question is concerned, the limited time constraint for this rebuttal prevents us to compare our method with state-of-the-art homotopy continuation method.
>
> If we correctly understood the reviewer's suggestion, we could proceed as follows.
>
> Consider the problem (3) from the paper and directly compute the gradient of the Lagrangian, and then find its roots.
> The main advantage of this method is that we do not need to diagonalize the matrix A.
> This gives the following system of nonlinear equations to solve (for $x \in \mathbb{R}^n$ and $\mu$ scalar):
>
>
> $$ 2(x - x^0) + \mu  (2 Ax + b) = 0    \quad \text{(A1)},$$
> $$ x^T A x + b^T x + c = 0		  \quad	\text{(A2)}.$$
>
> Let $y = (x, 0)^T$, solving (A1)-(A2) is equivalent to solving $F(y) = 0$, for an appropriate vector-valued function $F$.
>
> As we found no (easily installable) python library implementing homotopy continuation method, we decided to resort to scipy.optimize.fsolve.
>
> Concerning the execution time, the (new) Figure 1 shows that our algorithm is faster than fsolve for dimensions larger than 100.
>
> Moreover, this other method has the following drawbacks:
>
> - The method may converge to another root of (A1)-(A2) that is not the main solution of the projection (*i.e.*, a critical point that is not the global minimizer).
>
> - We have to choose a suitable starting point to ensure convergence. We decided to choose $[x^0 ; 0]$ as starting point (*i.e.*, $y^0 = (x^0, 0)^T$): it seems that we recover the correct solution (that is, the root of (A1)-(A2) corresponding to the optimal solution of (3)). If this choice seems reasonable, we have no theoretical guarantee.
>
> - In the degenerate cases, we still need to check the additional solutions (denoted as $x^\textrm{d}_k$).
>
> This last point is particularly troublesome because detecting that the case is degenerate (and computing these additional solutions) requires the eigendecomposition of A. This check is not considered in the execution time in Figure 1. We should add up the time of using fsolve (dashed orange line with star marker) and the time of the eigendecomposition (solid blue line with triangle down).
>
> Using fsolve (or another solver of systems of nonlinear equations) is therefore slower, has no guarantee of convergence, and may return an incorrect solution (*i.e.*, not the optimal minimizer).

---

### Official Review · Reviewer_WidE · 2022-07-09

**Rating:** 3
**Confidence:** 4
**Soundness:** 2 fair
**Presentation:** 3 good
**Contribution:** 1 poor

**Summary:**

This paper an introduction to a Python package implementing the method of projecting onto a central and non-cylindrical quadratic hypersurface (quadric) proposed in a recently posted arXiv manuscript [21]. Orthogonal projection onto a set is an important tool in machine learning as it is a basic building block of many learning algorithms. When the set onto which the projection is made is compact and convex the problem is relatively easy in the sense that the unique closest point exists and this point can be found using standard convex optimization algorithms. When the set is nonconvex, however, the problem needs to be tackled case by case. A quadric is a set of roots of a matrix quadratic equation characterized by the eigenvalues of the coefficient matrix $A$ of the quadratic term and the discriminant of the equation. This set is in general nonconvex. The recent manuscript [21, Proposition 2.20] provides  characteristics of a projected point (there can be multiple closest points from the input point since the set is nonconvex)  when the quadric is central (discriminant is nonzero) and non-cylindrical ($A$ is nonsingular), and also an algorithm [21, Algorithm 2] for computing it. This paper implements [21, Algorithm 2] in Python and build a package quadproj in order to democratize the algorithm. A review of the theory of [21] and exposition of how to use the quadproj package, with sample code snippets, are provided.

**Questions:**

None.

**Limitations:**

Please see "Strengths And Weaknesses" section. In the present form, this paper looks as if it is an appendix to [21].

**Strengths And Weaknesses:**

A strength of the paper is in the "democratization" of the algorithm for projection onto a quadric by writing an easy-to-use Python package. The package is designed so that construction of a quadric object, projection onto the quadric, and visualization of the solution is straightforward. The sample code snippets are also easy to follow. Indeed, visualization (in 2D or 3D cases)  is emphasized with a couple of figures. Also, the exposition is quite clear.

A weakness is in novelty. All the theory and method in Sections 2 and 3 are treated in [21], so the contribution of this paper seems to be limited to Section 4, description of the quadproj package. I understand that careful and easy-to-use implementation of an algorithm is an important task, but for such an implementation is publishable to a premire conference and journal, I think more consideration is needed.  For example, in the editorial policy of the SIAM Journal on Scientific Computing (https://epubs.siam.org/journal/sisc/editorial-policy), which I believe is closest in scope with the paper, states for the "Software and High-Performance Computing" category as follows.

> ... For software in particular, submitted papers should not be limited to describing a new package but must present the algorithmic or technological innovations that made the development of the software possible.

Unfortunately, I do not see either algorithmic or technological innovation in the exposition of the package in this paper.

---

> ### Author Response · Authors · 2022-08-01
> **Response to WidE**
>
> This reviewer does not ask any questions but raises the novelty as a weakness.
>
> We would like to point out that the NeurIPS Call For Papers (https://nips.cc/Conferences/2022/CallForPapers) markedly departs from the SISC editorial policy by explicitly mentioning "implementations" and "libraries". This policy motivated our submission to this conference.

---

> > ### Comment · Reviewer_WidE · 2022-08-09
> > **call for papers**
> >
> > I do not think NeurIPS is markedly different from the SISC in expectations. The premise of the cited Call For Papers is "We invite submissions presenting new and original research". I already mentioned a concern on novelty.
> >
> > If the paper would have been presented as suggested by Reviewer o4hv, or the library was written in such a way that overcomes limitations of the numpy or spicy with a good documentation, then that would be a welcome contribution to the community. I believe in the current form the paper does not fall into either category.

---

### Official Review · Reviewer_47QC · 2022-07-12

**Rating:** 7
**Confidence:** 3
**Soundness:** 3 good
**Presentation:** 4 excellent
**Contribution:** 3 good

**Summary:**

The paper describes a python package for projecting onto quadratic hypersurfaces. The basic problem it solves is to minimizing projection errors in Hilbert space subject to x^T A x + b^T x + c = 0. The authors show that with eigen decomposition of A, we can re-parametrize this problem and solve it with Newton's method. The paper also provides a number of example codes illustrating how to use the package.

**Questions:**

Could you provide some studies about how this package tool scales with the growth of problem size.

I would also like to see some high dimension application of the package.

**Strengths And Weaknesses:**

Strength

* clarity of writing
* the tool seems to have broader usage in practice

Weaknesses

* Some computational performance studies are missing. It would be nice to include statistics of how this package tool scales with the growth of problem size.

---

> ### Author Response · Authors · 2022-08-01
> **Response to 47QC**
>
> The reviewer mentioned the lack of computational performance study as a weakness.
> The reason for this lack of study is twofold.
>
> - On the one hand, we were limited by the nine content pages.
> - On the other hand, because the main part of the execution time is spent in the eigendecomposition that is used for diagonalizing the matrix $A$.
> This point is raised at the bottom of section 3: "These computations are negligible with respect to the eigendecomposition, which is the bottleneck of the method. In particular, for 100 problems of size n = 500, we obtain a mean execution time of 0.065 s for the root-finding algorithm and a mean execution time of 0.66 s for the eigendecomposition".
>
> Studying the (total) computation time of our package for large dimensions is equivalent to studying the computational time of the numpy.linalg.eig function.
>
> As a way to support our claim, we added a small experiment (Figure 1 in the new version of the paper, see highlighted text at the end of section 3) that shows how the execution time evolves with $n$, but we decoupled the time needed for diagonalizing the matrix with the time needed for the projection *per se*.

---

### Meta-Review · Area_Chair_hwyX · 2022-08-24

**Recommendation:** Reject
**Confidence:** Certain

**Metareview:**

The paper presents a software package to do projections on the non-cylindrical central quadratic hypersurfaces. While the problem is certainly interesting (all the reviewers agree), its motivation in the context of machine learning seems to be lacking in the paper. This is missing in the paper currently and is the main source of confusion in the reviewers' and the AC's minds. After discussions among the reviewers, I believe, the paper has much scope for improvements notwithstanding the merits. Please look at the suggestions carefully. Also, the paper, as it is, seems to better fit the scope of the MLOSS journal rather than the NeurIPS conference, just a thought from the AC. Having said that, I would encourage the authors to continue the development of this package.

**Award:**

No

---

### Decision · Program_Chairs · 2022-09-14

Reject